# Linear Dichroism Measurements for the Study of Protein-DNA Interactions

**DOI:** 10.3390/ijms242216092

**Published:** 2023-11-08

**Authors:** Masayuki Takahashi, Bengt Norden

**Affiliations:** 1School of Life Science and Technology, Tokyo Institute of Technology, Oookayama, Meguro, Tokyo 152-8550, Japan; 2Department of Chemical and Biological Engineering, Chemistry, Chalmers University of Technology, 412 96 Gothenburg, Sweden; norden@chalmers.se

**Keywords:** linear dichroism (LD), DNA/protein complex, RecA, Rad51, homologous recombination, cyclic AMP receptor protein (CRP), catabolism activator protein (CAP), transcription regulation, UvrB nucleotide excision repair protein, restriction enzyme

## Abstract

Linear dichroism (LD) is a differential polarized light absorption spectroscopy used for studying filamentous molecules such as DNA and protein filaments. In this study, we review the applications of LD for the analysis of DNA-protein interactions. LD signals can be measured in a solution by aligning the sample using flow-induced shear force or a strong electric field. The signal generated is related to the local orientation of chromophores, such as DNA bases, relative to the filament axis. LD can thus assess the tilt and roll of DNA bases and distinguish intercalating from groove-binding ligands. The intensity of the LD signal depends upon the degree of macroscopic orientation. Therefore, DNA shortening and bending can be detected by a decrease in LD signal intensity. As examples of LD applications, we present a kinetic study of DNA digestion by restriction enzymes and structural analyses of homologous recombination intermediates, i.e., RecA and Rad51 recombinase complexes with single-stranded DNA. LD shows that the DNA bases in these complexes are preferentially oriented perpendicular to the filament axis only in the presence of activators, suggesting the importance of organized base orientation for the reaction. LD measurements detect DNA bending by the CRP transcription activator protein, as well as by the UvrB DNA repair protein. LD can thus provide information about the structures of protein-DNA complexes under various conditions and in real time.

## 1. Introduction

Linear dichroism (LD), a polarized light spectroscopy method, is useful for studying filamentous molecules, such as DNA and protein filaments. These molecules can be easily macroscopically oriented by shear flow or electric fields [1,2,3]. Since long filamentous molecules are generally difficult to study by X-ray crystallography or NMR [2], LD can provide important complementary information about their structures in solution, including information about reaction mechanisms through time-resolved measurements. LD has long been considered a rather exotic technique, which may explain why it is not used more frequently. However, it has advantages over techniques. For instance, it can be readily performed in solution under variable conditions using current commercially available devices. In this review, we highlight the applications of LD in the analysis of DNA-protein interactions, with the anticipation that this could lead to its increased use. We first describe the basic principles of LD and then present some examples of its applications.

## 2. Principles and Measurements

### 2.1. Principle of Linear Dichroism

LD measures the differential absorption of orthogonal forms of linearly polarized light [1,2,3]. Chromophores absorb UV/visible light, polarized in specific directions, according to their electric dipole transition moments (Figure 1a). Therefore, when all molecules are perfectly aligned, the absorption of polarized light will be at its maximum when the electric field is parallel to the transition moment. The difference between the absorption of light polarized parallel and perpendicular to a macroscopic (laboratory) axis is denoted as LD = Abs_parallel_ − Abs_perpendicular_ [1,2,3]. In addition to chromophore orientation, the LD signal intensity linearly depends on the optical pathway, sample concentration, and the absorption coefficient of the chromophore. Therefore, the ‘reduced’ LD (LD^r^ = LD/isotropic absorbance) is independent of these parameters and only depends on the macroscopic molecular orientation, thus reflecting the transition moment orientation. In addition to the local orientation of the transition moment, LD^r^ scales with the macroscopic alignment of sample molecules, which is denoted by the orientation factor S (0 ≤ S ≤ 1):LD^r^ = S × 3/2 × (3 × <cos^2^Θ> − 1)(1)

S is 1 for perfect alignment and 0 for random orientation. Equation (1) contains two unknown parameters, S and Θ, which require extra information. The orientation factor, S, may, for example, be determined from small-angle X-ray scattering (SAXS) or small-angle neutron scattering (SANS) measurements under identical alignment conditions [4].

Since each chromophore generally absorbs light at different wavelengths, and due to transition moments at different angles within a specific plane for an aromatic chromophore (Figure 1a) [1,2], chromophore orientation can be assessed from the LD signal measured at corresponding wavelengths and related to the respective orientations of transition moments. The spectral shape of LD is generally independent of the orientation factor, although the LD amplitude is scaled by it, and is obviously 0 for random orientation [1,2]. Therefore, in favorable cases, chromophore orientation can be determined without the need to independently determine the orientation factor.

The orientation factor itself provides information about the size, shape, and stiffness of molecules because the signal intensity depends on the alignment degree of the sample molecules according to Equation (1). Stiff, long, and straight molecules produce a maximum signal (large S), while short, flexible, or bent molecules produce a weak signal (small S).

### 2.2. Utility and Advantages of LD Measurements

#### Detection of Changes in Shape and Flexibility of DNA

Long DNA align better than short DNA fragments and produce a strong LD signal. Therefore, DNA digestion by endonucleases can be measured in situ by observing a decrease in LD signal intensity (Figure 2) [5,6]. In this way, LD allows for continuous real-time observation of DNA digestion.

The degree of alignment of DNA fragments (S) also depends on their straightness and stiffness. Straight DNA aligns better than curved DNA. Thus, it is possible to detect the bending of DNA caused by proteins [7,8]. Protein binding to DNA usually stiffens the DNA, which facilitates the alignment of DNA and increases LD signal intensity. The stiffening of DNA by RecA and Rad51 recombinases and the glucocorticoid receptor protein has been observed using LD [9,10,11]. The increase in LD signal intensity has been used to study the DNA binding of RecA and Rad51 [5,9].

### 2.3. Binding Mode (Intercalation or Groove Binding)

LD can determine the binding mode of ligands to DNA, which is essential for understanding various mechanisms, including those related to medicinal drug action. In the case of intercalation, the orientations of the ligand and the DNA bases are expected to be the same, with both having molecular (aromatic) planes approximately perpendicular to the DNA helix axis in B conformation DNA (Figure 3). In the case of groove binding, the ligand plane is not perpendicular to the DNA helix axis but is generally parallel to the groove direction (45° to the helix axis for the minor groove) [1,2,12,13,14]. When we insert Θ = 90° into Equation (1), corresponding to intercalation, it gives a strong negative LD value, similar to that observed for transition moments in the likewise perpendicular nucleobases. However, with Θ = 45°, a positive LD occurs with a weaker magnitude. It is important to note that <cos^2^Θ> means that we are dealing with an ensemble average over a potentially broad distribution over different angles, so caution must be exercised when interpreting the data. Due to some dynamics for DNA in solution, Θ, as determined from Equation (1), is thus often less than 90°, but the same ‘effective’ angle is generally found for an intercalation molecule, as if it is moving in synchrony with the bases. In Equation (1), LD vanishes for Θ = 54.7°, and this structure cannot be distinguished from a case of isotropic orientation, where LD vanishes due to S = 0. For this reason, it is often advisable to get information about S from an independent source, such as SANS [4].

By exploiting the fact that there are transition moments with different directions in nucleic acid base pairs, LD can be used to determine the tilt and roll angles of DNA bases, thus distinguishing A-form from B-form DNA [1,2,16]. LD measurements have demonstrated that the binding of RecA and Rad51 recombinases promotes a perpendicular orientation of the DNA bases of single-stranded DNA (ssDNA) [4,9,17,18]. In the absence of protein, the DNA bases of free ssDNA move relatively freely and do not exhibit any preferential orientation (extremely small LD). This structural change, a reflection of how the protein steers the orientation of the nucleic bases, is probably essential for the strand exchange activity of these proteins.

### 2.4. Advantages of LD Measurements

One of the advantages of LD is that it can be used to perform structural measurements in solution under various solvent conditions. Thus, the structural modifications of molecules caused by effectors and changes in solvent conditions (salts, pH, temperature, etc.) can be examined by LD. In contrast to another type of polarized spectroscopy, circular dichroism (CD), which provides information in terms of relative chromophore-chromophore orientation, LD provides information about the orientation of individual chromophores. The analysis is also more straightforward, although the two methods are complementary [1,2].

Since small molecules do not become flow-oriented and therefore do not produce any LD signal, the LD signal of the complex can be selectively measured, even in the presence of excess non-bound ligands or proteins. Furthermore, LD is a sensitive method that can be performed in relatively dilute solutions, allowing for good signals to be obtained in the µM (in bases) range of DNA concentration. The sample can be aligned in a commercialized mini-Couette cell with a volume of less than 100 µL [19].

### 2.5. Difficulties in the Analysis of LD Signals

One of the main difficulties in analyzing LD signals of DNA-protein complexes lies in the presence of several chromophores simultaneously absorbing light in the same wavelength region. The absorption bands of DNA bases and the aromatic residues (Tyr and Trp) of proteins overlap. Furthermore, because identical chromophores (Trp and Tyr residues) can occupy different positions within a protein, it is challenging to get information about one specific chromophore within a complex. One approach is to replace a target chromophore with a structurally similar chromophore that absorbs light differetly. Changes in LD spectra after such chromophore replacements can provide valuable information about the target chromophore. Another approach is to detect the LD signal through fluorescence, since fluorescence measurements tend to be more selective.

Another difficulty and associated ambiguity, as mentioned above, is that the orientation factor must be determined for angular analysis. The orientation factor can be estimated by performing SAXS (or SANS) scattering and LD measurements on the same sample [4]. However, this method consumes a large amount of materials. Furthermore, the orientation factor may change upon complex formation. For instance, the orientation factor of a complex with one protein differs from that of a complex with two proteins. One must verify the binding stoichiometry using independent measurements [9].

## 3. Measurements

The LD signal (absorption difference) can be measured with high sensitivity using a CD spectrometer adapted to work in LD mode. Recent CD spectrometers can perform measurements in LD detection mode. LD signals can also be detected by measuring fluorescence instead of absorption [2,19,20,21,22], which allows for the selective analysis of fluorescent chromophores in the sample. It can even be used for single-molecule observations [22].

## 4. Methods of Sample Alignment

### 4.1. Couette Cells

A Couette cell is comprised of two co-axial cylinders with a small gap (Figure 4). The sample solution is introduced into this gap. Rotating one of the cylinders creates a shear force that aligns the elongated sample molecules. The LD signal can be continuously observed, allowing measurements throughout the entire spectrum (signal variation with wavelengths), including real-time kinetic analysis. However, this method can only align relatively long DNA (>2000 bp long). The shear force can be changed by changing the rotation speed to observe the effects of different shear forces, which provides further information about the stiffness and straightness of the molecules. Many of the LD analyses presented below were performed using Couette cells.

### 4.2. Flow Cells

The shear force can also be created by passing the sample solution through a thin flow cell (Figure 4). However, the shear force generated in this way is usually weaker than that produced by a Couette cell. It varies over the cross-section of the thin flow cell, whereas it is constant in a Couette flow cell. The solution is circulated using a pulsation-free peristaltic pomp. The volume of sample solution required for this system is larger than that required for a mini-Couette cell. In contrast, this cell type is free of optical defects caused by the strain birefringence of thick cylindrical silica windows and generally provides a better baseline extending into the far UV. The cell is also better adapted for fluorescence measurements. Another advantage is that the flow direction can be changed by rotating the cell, which facilitates signal analyses [23]. 

### 4.3. Application of Electric or Magnetic Field

A high electric field can align even short DNA fragments (43 base pairs long) [24,25]. However, a high electric field can only be applied briefly (as a pulse). The sample molecules are aligned for a short time and then disoriented. Although the entire LD spectrum cannot be measured using this method, the time required for the loss of orientation after pulse termination can be measured, which provides information about the shape of the DNA (length and straightness). This method can detect DNA bending promoted by protein-binding or DNA-bending sequences [7,24,25,26]. However, it is limited because it can only be performed under very low salt conditions.

Applying a weaker electric field promotes the electrophoretic movement of DNA in electrophoretic gels, which can be used to align long DNA molecules due to the mechanical interaction between the DNA and the gel fibers [27]. Alignment of single-stranded DNA has been reported using a very strong magnetic field [28].

### 4.4. Deformation of Sample-Loaded Film or Gel

One method to align small molecules is to deposit the sample on a film or a polyacrylamide gel and stretch the film or deform the gel. This technique can align small molecules that cannot be aligned using other methods [29,30,31].

## 5. Examples of LD Applications

### 5.1. Determination of the Binding Mode of Ligands (Intercalation or Groove Binding)

LD signals provide information about the orientation of chromophores relative to the filament axis. Intercalating ligands are oriented parallel to DNA bases, while groove-binding ligands fit into the DNA groove, resulting in their orientation differing from that of the DNA bases (Figure 3). In an ideal situation, ligands would absorb light in wavelength regions different from those of DNA bases, which would allow the ligand orientation to be selectively probed by LD [12,13,14]. It is important to note that there can be potential ambiguity due to the ensemble average, <cos^2^Θ>, as mentioned earlier.

### 5.2. Kinetic Analysis of DNA Digestion by Endonucleases

A real-time measurement of DNA digestion by endonucleases, including restriction enzymes, is difficult and cannot be done using conventional spectroscopy (absorption, fluorescence, CD). Usually, digestion is analyzed by incubating the DNA with the restriction endonuclease, removing aliquots of the reactions at different times, and separating the DNA fragments through gel electrophoresis, which is time-consuming and laborious.

LD is sensitive to changes in DNA length, because short DNA is less well-oriented and exhibits a weaker LD signal than long DNA. The reaction can be followed in real-time and under various conditions (Figure 2) [5,6,32], which facilitates the analysis of digestion kinetics. A quantitative analysis only requires the pre-calibration of the LD signal as a function of fragment length.

### 5.3. DNA Bending by UvrB DNA Repair Protein

UV-light damages DNA by forming pyrimidine dimers. The damage is mainly repaired by the UvrABC exonuclease complex in *E. coli* [33]. The UvrA dimer forms a complex with the UvrB dimer to recognize the UV-damaged DNA. The UvrA dimer dissociates from the DNA, and the UvrB dimer recruits the UvrC exonuclease, which cuts the DNA on both sides of the lesion to remove the damaged region. The UvrABC exonuclease also repairs other DNA adducts [33]. However, precisely how the UvrA_2_UvrB_2_ complex recognizes UV-light damaged DNA has remained unclear.

Structural changes that occur in DNA upon binding of the UvrA_2_UvrB_2_ complex have been analyzed using flow LD [8,34]. UV damage leads to a slight decrease in the LD signal intensity of DNA, suggesting that the flexibility of the DNA is slightly increased due to UV-induced damage. The LD signal is drastically reduced when damaged DNA is incubated with UvrA and UvrB, suggesting that wrapping of the DNA occurs around UvrB_2_ [8]. This change is specific to UV-damaged DNA.

Interestingly, a UvrB mutant that can bind to UV-damaged DNA but cannot recruit UvrC does not significantly affect the LD signal intensity [34]. A footprint analysis showed that the mutant does not kink the DNA, which is essential for the binding and incision of the DNA by UvrC. This structure was later confirmed through crystallographic analysis of the complex [35].

### 5.4. DNA Bending by CRP Transcription Activator

Regulation of gene expression is an essential feature of all living organisms. Some genes require one or more transcription activator proteins for efficient transcription. CRP (or CAP) activates the transcription of several genes involved in sugar metabolism, such as β-galactosidase in *E. coli* [36,37]. It binds near the promoter region of these genes in a manner reliant on the nucleotide sequence to facilitate their transcription. One unanswered question is how the binding of CRP activates transcription. A previous study reported that the CRP/DNA complex has abnormal electrophoretic mobility, suggesting DNA bending [38,39].

The deformation of DNA upon CRP binding was determined by measuring the degrees of orientation and disorientation times of the complexes using electric pulse LD [7]. By applying a high electric field pulse to the solution, short DNA fragments were aligned with and without CRP. The complex was then oriented for a short pulse time and thermally disoriented afterwards. The disorientation time became shorter upon CRP binding [7]. Simulation analysis indicated a large bend of 180 degrees (hairpin fold).

Later crystallographic analysis of the complex confirmed the presence of significant DNA bending (kinking) [40], although not to the same degree as estimated by LD. This discrepancy may be due to the shorter length of the oligonucleotide used in crystallographic analysis [40,41]. Porshcke and collaborators performed LD measurements with the DNA fragment used in the crystallographic analysis and observed similar bending [41]. Thus, it appears that the DNA length used in crystallography was insufficient to maintain extensive bending of the DNA. LD measurements can be performed using various substrates and under various conditions, providing unique information that is difficult to obtain through conventional structural analyses.

Later analyses showed that various transcription activators, including the TATA box-binding protein, induce DNA bending [42,43]. DNA bending facilitates the binding of RNA polymerase to promoters and the opening of DNA for transcription [43].

### 5.5. Structure of Single-Stranded DNA in Complexes with RecA and Rad51 Recombinases

RecA and Rad51 recombinases promote DNA strand exchange between two DNAs with identical or similar nucleotide sequences for homologous recombination [44,45]. They participate in the repair of DNA double-strand breaks and the regression of stalled replication forks. Rad51 recombinase is also involved in chromosome pair formation during meiosis. Rad51 is essential for the survival of higher eukaryotes [46] and has been implicated in carcinogenesis and cancer progression [47]. Rad51 is, therefore, a potential target for new cancer treatments [47,48,49], but the development of such treatments depends on a deeper understanding of the reaction mechanisms and the detailed structures of recombination intermediates.

Biochemical analysis has shown that these recombinases first bind to single-stranded regions of DNA with high cooperativity, subsequently forming a helical filament around the DNA [50]. They then bind to double-stranded DNA with a sequence similar to that of the single-stranded DNA and perform a strand exchange reaction in the presence of ATP [51,52,53]. Since all these processes occur within the filament, determining the filament structure is essential for understanding the reaction mechanism. However, using conventional methods to determine the structure of such long DNA/protein filaments is difficult. This has incited strong interest in developing and using LD methods to determine these structures [4,5,10,17,18,20].

### 5.6. Stiffening of Single-Stranded DNA by RecA and Rad51 Recombinases

Single-stranded DNA (ssDNA) is flexible, with its nucleobases moving freely because of the absence of DNA base pair formation and strong stacking. Thus, ssDNA does not exhibit any significant flow LD signal. However, the binding of RecA or Rad51 to ssDNA dramatically increases the LD signal [4,9,10,17,18]. The LD signal intensity is strong even when weak shear forces are applied, indicating the remarkable straightening of the DNA upon protein binding. Electron microscopy observations have confirmed the stiffening and straightening of ssDNA upon the binding of RecA or Rad51 [4]. This stiffening effect may facilitate the docking and matching of ssDNA with a second DNA molecule.

### 5.7. Immobilization of Nucleobases of Single-Stranded DNA in the Presence of Strand Exchange Activating Agents

In the case of the RecA/ssDNA complex, the spectral shape is modified by the addition of ATP (or its non-hydrolyzable analog ATPγS) [9]. RecA requires ATP for its strand exchange activity [51,52]. In the presence of its analog, ATPγS, the spectra show a negative band around 260 nm (Figure 5), while only a positive band around 280 nm is observed in the absence of ATP [9]. Since DNA bases mainly absorb UV light around 260 nm, the negative LD signal centered at 260 nm probably originates from DNA bases. This negative signal indicates a relatively perpendicular orientation of DNA bases to the filament axis. Without ATP, the DNA bases would be mobile and not produce any LD signal. The positive signal around 280 nm is probably attributed to the aromatic residues (Trp, Tyr) of RecA, which have UV absorption centered around 280 nm.

The strand exchange activity of Rad51 is weaker than that of RecA and requires Ca^2+^ [54] or the Swi5/Sfr1 protein [55] in addition to ATP, for optimal activity. Interestingly, the LD spectra of the Rad51/ssDNA complex exhibit a positive band around 280 nm and a weak negative band around 260 nm in the presence of ATP [17,18]. Adding Ca^2+^ or the Swi5/Sfr1 activating protein promotes the appearance of a sizeable negative band around 260 nm [17,18], which corresponds to the perpendicular orientation of DNA bases.

### 5.8. Perpendicular Orientation of DNA Bases in RecA and Rad51 Filaments

Because of the partial overlap of UV absorptions of DNA, ATP, and protein aromatic residues (Trp and Tyr), two approaches have been used to prove that the negative band around 260 nm is indeed produced by DNA bases:(a)A DNA analog (poly(dεA)) which absorbs UV light above 310 nm at a wavelength at where no other reaction constituents absorb light, was used [4,55] to unambiguously determine the orientation of poly(dεA) bases. Indeed, as expected, a negative LD signal was observed above 310 nm in the presence of ATPγS, supporting the perpendicular orientation of DNA bases in the recombinase DNA/complex (Figure 6) [9,55].(b)A genetically engineered RecA protein was employed, in which one of its two Trp residues, the primary UV-absorbing residues, was replaced with a structurally similar residue (Tyr or His) with weaker UV-absorbance [56,57]. The poly(dεA)-engineered RecA complex also produced a negative LD signal above 310 nm, that was identical to that of the complex formed with wild-type RecA (Figure 7). However, as expected, there was a decrease in the positive band around 280 nm (Figure 7), indicating that the positive signal was produced by the Trp residues of RecA and not by the DNA bases, thus confirming the perpendicular orientation of the DNA bases in the active complex. This conclusion was later confirmed by crystallographic and cryo-EM analyses [58,59].

In contrast to the complex formed in the presence of ATPγS, only a positive LD band centered around 280 nm was observed with poly(dεA) in the absence of ATPγS, suggesting no significant perpendicular orientation of DNA bases.

Similar studies conducted with Rad51 [17,18,60] also demonstrated the perpendicular orientation of ssDNA bases in the Rad51 filament in the presence of Ca^2+^ ions or Swi5/Sfr1, which was later confirmed by cryo-EM analysis [61].

These results indicate that the ssDNA structure in the active RecA or active Rad51 complex is similar to that of B-like dsDNA with perpendicularly oriented bases. This well-organized structure is postulated to facilitate pairing with the complementary DNA strand, an essential step in the DNA strand exchange reaction.

The absence of measurable DNA base orientation, which indicates high base mobility in the absence of an activator, supports the importance of the stiff, perpendicular DNA base orientation for DNA recombination. X-ray crystallography cannot resolve the structure of the complex in the absence of activators (ATP for RecA and ATP and Ca^2+^ for Rad51), thus again demonstrating the utility of LD measurements.

### 5.9. Building a Molecular Model of RecA/DNA and Rad51/DNA Complexes

Crystallographic analysis of filamentous molecules is difficult. Determining the structure of the DNA-RecA complex took considerably more time (16 years) than the structural determination of RecA alone [58,59,62]. Chen and collaborators analyzed the structure of the DNA-RecA complex using covalently linked RecA oligomers [58]. A molecular simulation was required to determine the structure of the complex filament.

Molecular model building of RecA-DNA and Rad51-DNA complexes was performed using LD and RecA or Rad51 structures without DNA determined through X-ray-crystallographic analyses [62,63].

The orientation of the RecA protomer within the active filament was investigated by determining the orientation of aromatic residues in RecA using LD [57]. This was determined as follows: the LD signal of each aromatic residue was resolved by replacing the target residue (Tyr or Trp) with a structurally similar but non-UV-absorbing residue (Phe or His). The difference in LD signal intensity between the wild-type RecA and the engineered RecA provided the LD signal of the removed target aromatic residue. The orientation of these aromatic residues, which can be resolved in terms of two angles relative to the common fiber axis, indicates the orientation of each RecA protomer within the complex filament and provides information about the conformational changes occurring in RecA upon DNA binding [57]. A similar study was performed for the Rad51-DNA complex filament [60].

## 6. Conclusions and Future Directions

LD spectroscopy is a powerful technique for studying protein-DNA interactions in aqueous solution under a wide range of conditions. LD can thus be used to investigate structural changes in target molecules induced by regulatory elements and various environmental factors (salt, pH, temperature, etc.). Such investigations would be difficult, if not impossible, using other methods. LD can also be used for kinetic analyses of protein-DNA transactions, such as DNA digestion by endonucleases in real time. LD can also be used to study various types of DNA bending by transcription regulatory proteins or DNA repair enzymes. LD has been crucial in revealing the stiffening and straightening of single-stranded DNA by RecA and Rad51 recombinases, as well as the perpendicular orientation of DNA bases in RecA- and Rad51-DNA complexes. Notably, since LD can be performed under widely different conditions, it is possible to demonstrate that perpendicular base orientation only occurs in the presence of recombinase activators, indicating the importance of this structure in the strand exchange reaction.

In addition to studying DNA/protein interactions, LD can be used to analyze other filamentous molecules, such as β-amyloid and prion fibers, whose formations are associated with neurological diseases [2,23,64]. LD measurements can facilitate the analysis of fibril structure and its formation kinetics, which are essential for understanding amyloidosis diseases and developing treatments for these diseases. Recently, LD has been used to study ligand/cell membrane interactions [65,66,67,68,69,70], using cell membrane-mimicking liposomes aligned in a Couette cell. These studies on membrane structure would be difficult using other methods. The motion of chromophores in single molecules can be studied by fluorescence-detected LD. These measurements allow us to analyze molecular dynamics, such as DNA breathing [22]. LD observations of molecular assembly within living cells have also been reported [70,71,72,73].

## Figures and Tables

**Figure 1 ijms-24-16092-f001:**
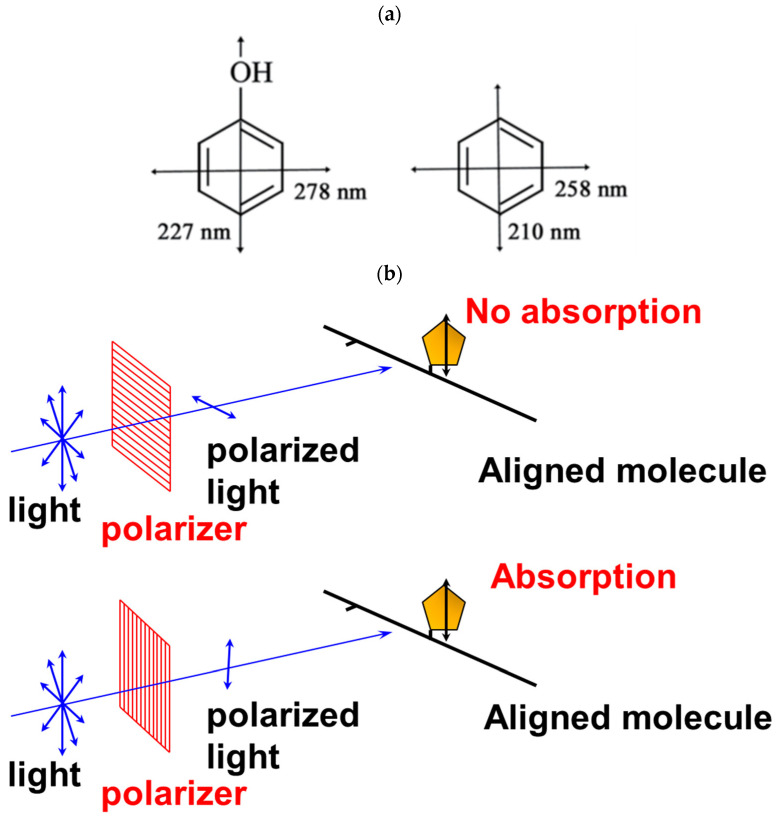
Transition dipole moments in a chromophore (**a**) and principle of linear dichroism (**b**).

**Figure 2 ijms-24-16092-f002:**
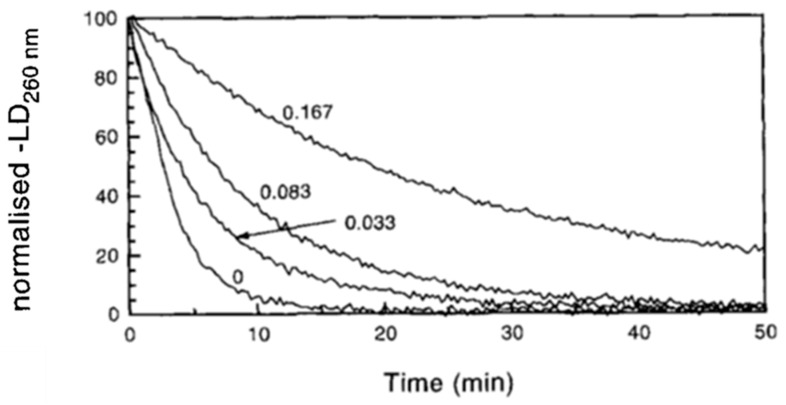
Linear dichroism observation of DNA digestion by endonuclease I in the presence of various concentrations of ethidium bromide. From [5].

**Figure 3 ijms-24-16092-f003:**
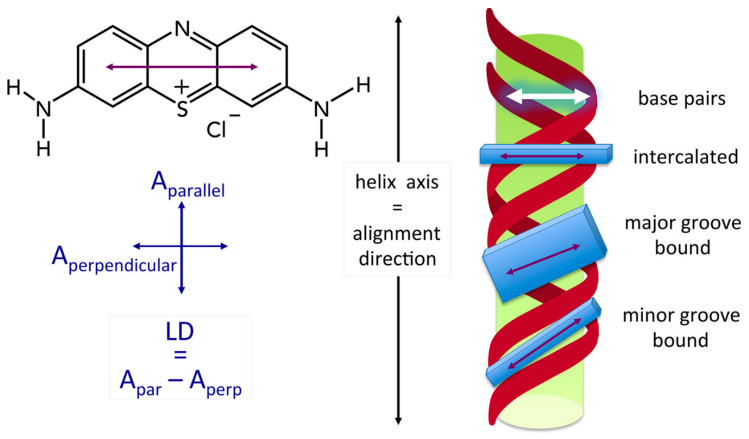
Schematic presentation of the geometry of groove binding and intercalation ligands. From [15].

**Figure 4 ijms-24-16092-f004:**
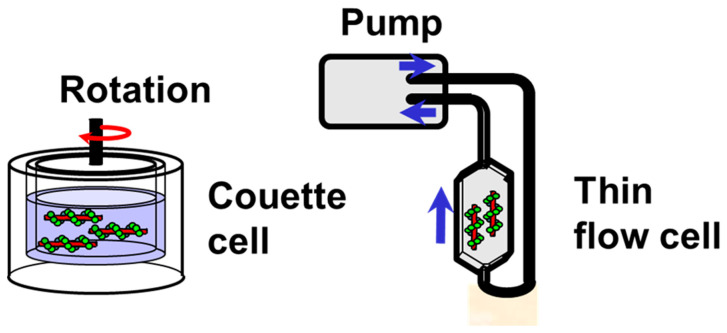
Couette cell and flow-through cell for LD measurements.

**Figure 5 ijms-24-16092-f005:**
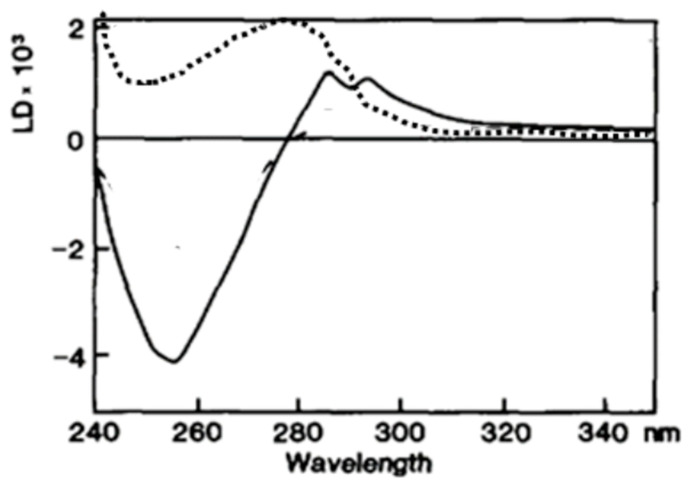
ATP cofactor dependence of RecA/ssDNA complex LD spectra. The LD spectrum of the RecA-ssDNA complex (dots) shows a positive signal around 280 nm, while that of the ATPγS-RecA-ssDNA complex shows a strong negative signal around 260 nm (continuous line). From [9].

**Figure 6 ijms-24-16092-f006:**
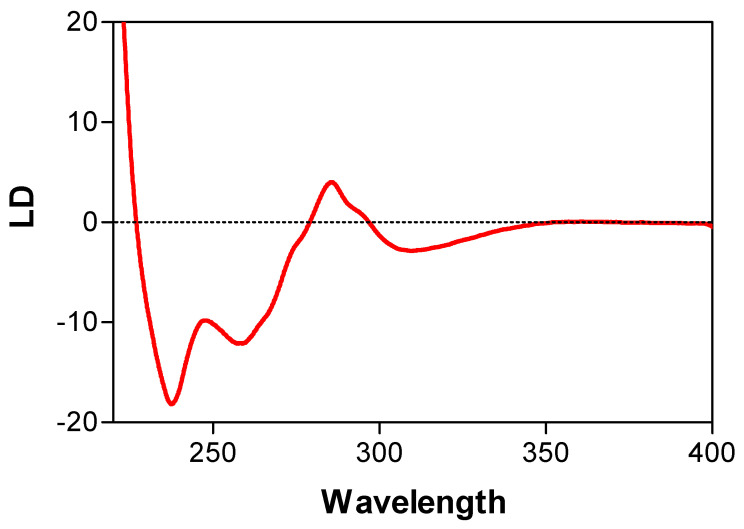
Use of poly(dεA) for assigning changes in the LD signal to the DNA bases in the RecA/ssDNA complex. Poly(dεA) absorbs light above 310 nm, while RecA does not. Furthermore, poly(dεA) absorbs strongly around 240 nm. The negative LD signals above 310 nm and around 240 nm indicate the perpendicular orientation of DNA bases in the complex. From [4] (Copyright Elsevier).

**Figure 7 ijms-24-16092-f007:**
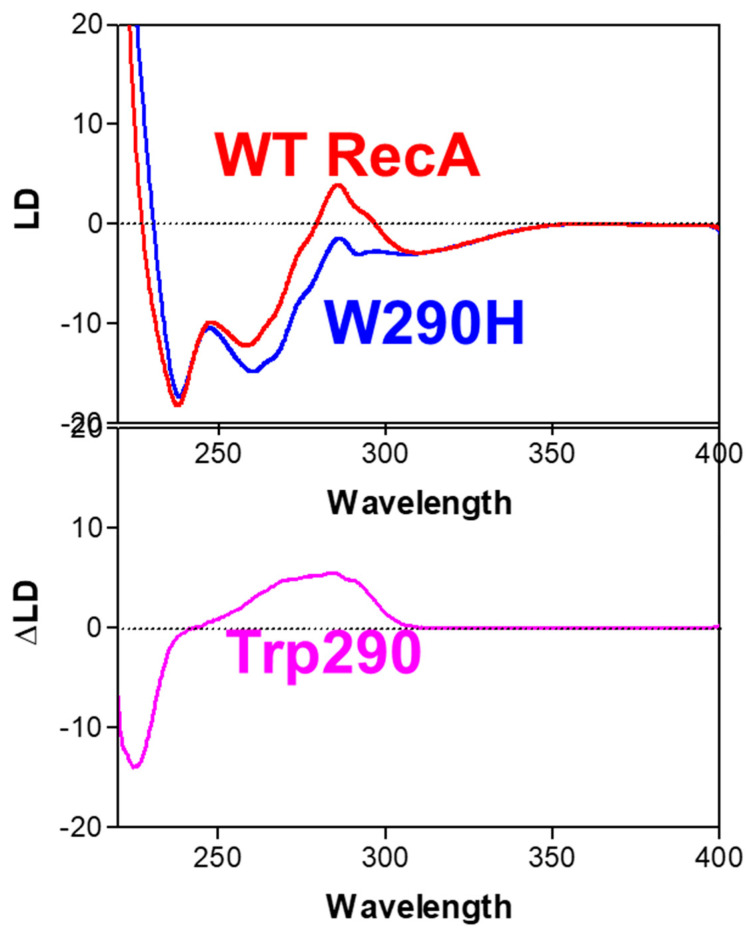
Determination of the orientation of one tryptophan residue of RecA in the RecA-ssDNA complex. The LD spectra of the complex formed with poly(dεA) and wild-type RecA, and that of the complex formed with poly(dεA) and engineered RecA, in which tryptophan 290 is replaced with histidine, were measured. The tryptophan 290 signal was estimated by subtracting the spectra with RecAW290H from those with wild-type RecA. From [56] (Copyright Elsevier).

## Data Availability

Not applicable.

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
