# Peer review of "Linear Dichroism Measurements for the Study of Protein-DNA Interactions"

_ijms, 2023, doi:10.3390/ijms242216092_

Round 1

Reviewer 1 Report

Comments and Suggestions for Authors

The author summarized the methodology, and application of using linear dichroism (LD) to study the protein-DNA interaction by taking advantage of the filamentous nature of DNA and the absorption anisotropy of the "ligands" in aligned samples. The article is easy to understand with clear delivery of the concept in the introduction followed with details of  sample preparation and examples of applications. 

However, it seems to me that their are drawbacks of this technique that needs to be discussed and the author should propose some solutions in order for the further advancement of this technique.

1. LD determines the shape change of DNA, binding mode of ligand to DNA through measuring the transition dipole moment of a chromophore that binds to DNA. However, as an ensemble measurement, the spectra one attains is essentially a result of a mixture of chromophores with different orientations. Therefore, one gets the information of orientation distribution of the chromophores that binds DNAs instead of absolute orientation. It will be helpful if the authors can write a paragraph with a detailed example showing the workflow of data analysis and calculate the orientation distribution based of their LD data.

2. Following the previous comments, as an ensemble measurement on an aligned sample, sample alignment also become important. The author listed three alignment methods. However, how does one determine the quality of the alignment if they do not use LD? The information on the alignment without chromophores binding to the sample is important because the orientation factor S is convoluted with chromophore binding orientation and the orientation distribution of DNAs, one has to know both to give a solid conclusion on DNA-Protein binding.  For instance, if the chromophore is bound 45 degree to the DNA, however the alignment of DNA results in DNAs parallel to the sample cell but DNAs can either be 0 degree or 180 degree to the sample cell. In this case one would concluded that the binding of the chromophore is random since LD will be 0, but the actual binding is not random. Of course one can pull out the raw data of Abs(Parallel) and Abs(Perpendicular) and tell that it's not random, but this is just one of the simplest situation. In reality, things can be much more complicated.

3. On the sample alignment section. It may not be clear to the reader how a Couette cell aligns the sample, it seems that the orientation of the alignment depends on the sample position relative to the cell, since the shear force is tangent to the curvature.

4. The author mentioned using magnetic and electric pulses to align the sample, which suffers from short alignment lifetime. Can this issue be avoided by using static electric or magnetic field?

5. Lastly, the author gave a couple of examples of LD application. However, those examples seemed to suffer from the issues I listed above. I wonder if authors can write in details on the data analysis to separate different orientation distributions. 

5.1. In DNA digestion kinetics, the author claimed that real-time kinetic measurement is not possible with absorption, fluorescence or CD but did not explain why that is the case. Also, MALDI-TOF seems to be a perfect method for the purpose.

5.2. ssDNA stiffening. The author should provide a separate evidence, e.g. CryoEM data or something else to support that the absence of LD signal is due to the flexibility of ssDNA in the flow cell. 

Minor issues.

1.The formats of the references are not consistent. Some of them have URLs others don't. Please refer to the journal guidelines on the formatting.

2. Figure 1: The caption should be transition dipole moments instead of transition moments. in Figure 1a, it is not clear why the number 3 and 4 are present.

3. Figure 4: In the caption the number "4" is in a smaller font size than others.

Comments on the Quality of English Language

The English language needs to be improved significantly.

Example:

1. Abstract: "On the other hand, the signal intensity also depends upon the degree of sample alignment". Here also is redundant. the degree of sample alignment can be changed to the alignment. "Therefore, LD can detect DNA shortening, bending here...." Here LD does not directly detect these effect since LD is not a detection method.

2. Principle of Linear Dichroism: "Therefore, if all molecules are aligned in one direction, the absorption of polarized light will be maximum when parallel to the transition moment." The second part of the sentence is misleading, because the author actually meant the absorption will be at maximum when the incident light polarization is parallel to the transition dipole moment of the molecule.

"The signal intensity depends, in addition to the chromophore orientation, on the concentration and the absorption coefficient of the chromophore. Therefore, the reduced LD (LDr = LD / isotropic absorbance), is independent of these parameters and conveniently used to analyze the transition moment orientation" The sentence after "Therefore" is not the cause of the sentence before "therefore".

"Since each aromatic chromophore absorbs light at several wavelengths and in a specific plane (several transition moments)" The word Aromatic chromophore is ill-defined and is seldom used in scientific publication. The whole sentence is not clear, although experienced readers can understand what they mean.

"the chromophore orientation can be determined by measuring LD at various wavelengths and determining the orientation of all transition moments." In this sentence word "determine" repeated twice and made it confusing on what determines what.

There are more language issues along the way that need to be fixed. 

Author Response

The author summarized the methodology, and application of using linear dichroism (LD) to study the protein-DNA interaction by taking advantage of the filamentous nature of DNA and the absorption anisotropy of the "ligands" in aligned samples. The article is easy to understand with clear delivery of the concept in the introduction followed with details of  sample preparation and examples of applications.

However, it seems to me that their are drawbacks of this technique that needs to be discussed and the author should propose some solutions in order for the further advancement of this technique.

  1. LD determines the shape change of DNA, binding mode of ligand to DNA through measuring the transition dipole moment of a chromophore that binds to DNA. However, as an ensemble measurement, the spectra one attains is essentially a result of a mixture of chromophores with different orientations. Therefore, one gets the information of orientation distribution of the chromophores that binds DNAs instead of absolute orientation. It will be helpful if the authors can write a paragraph with a detailed example showing the workflow of data analysis and calculate the orientation distribution based of their LD data.

We appreciate the comment. The ms has been modified to be more clear, see also below.

  1. Following the previous comments, as an ensemble measurement on an aligned sample, sample alignment also become important. The author listed three alignment methods. However, how does one determine the quality of the alignment if they do not use LD? The information on the alignment without chromophores binding to the sample is important because the orientation factor S is convoluted with chromophore binding orientation and the orientation distribution of DNAs, one has to know both to give a solid conclusion on DNA-Protein binding. For instance, if the chromophore is bound 45 degree to the DNA, however the alignment of DNA results in DNAs parallel to the sample cell but DNAs can either be 0 degree or 180 degree to the sample cell. In this case one would concluded that the binding of the chromophore is random since LD will be 0, but the actual binding is not random. Of course one can pull out the raw data of Abs(Parallel) and Abs(Perpendicular) and tell that it's not random, but this is just one of the simplest situation. In reality, things can be much more complicated.

We are grateful to the Reviewer for raising this point - and also the other related points that could lead to misunderstanding to a general audience. There is not space enough for a complete background (we give appropriate references) but it is, just as the Reviewer says, important to stress that we are not dealing with single molecules but ensembles, and furthermore also ensembles of orientations in each molecule due to thermal dynamics. In order to simplify we were apparently too simple and removed the average signs – they are now re-introduced in Eq (1) and we also introduce a note of caution to emphasize that we have a two-parameter problem (S and theta) and also the lack of information about the angular distribution function as we only determine <cos2 theta>.

  1. On the sample alignment section. It may not be clear to the reader how a Couette cell aligns the sample, it seems that the orientation of the alignment depends on the sample position relative to the cell, since the shear force is tangent to the curvature.

The Reviewer is (again) correct but we will refrain from going into details as there are other complications (due to that the S matrix diagonalizing angle – starting at exactly 45° - changes to 90° - in the flow cell frame, when S in Eq (1) goes from 0 to 1.  This has no effect on the interpretation of Eq (1) since the direction of optical incidence is radial (not axial) – again the reader is referred to the literature for details how to analyze the optical data.

  1. The author mentioned using magnetic and electric pulses to align the sample, which suffers from short alignment lifetime. Can this issue be avoided by using static electric or magnetic field?

We are thankful to the Reviewer for taking up this point. Yes, a very strong magnetic field could produce DNA orientation – but very weak typically estimated to S=0.000001 in a 70 T magnet. The small orientation and required big magnet makes it inconvenient though.

One cannot apply a high electric field for an extended period (longer than 1 ms) because of strong Joule dissipation (boiling the sample) and other effects destroying the long DNA molecules by ripping forces..

  1. Lastly, the author gave a couple of examples of LD application. However, those examples seemed to suffer from the issues I listed above. I wonder if authors can write in details on the data analysis to separate different orientation distributions.

We add some general comments in this regard.

5.1. In DNA digestion kinetics, the author claimed that real-time kinetic measurement is not possible with absorption, fluorescence or CD but did not explain why that is the case. Also, MALDI-TOF seems to be a perfect method for the purpose.

There is no change in absorption or CD signals upon DNA digestion because these signals reflect the base stacking. We cannot use fluorescence measurements because DNA is not fluorescent. MALDI-TOF measurements cannot follow the reaction continuously.

5.2. ssDNA stiffening. The author should provide a separate evidence, e.g. CryoEM data or something else to support that the absence of LD signal is due to the flexibility of ssDNA in the flow cell.

We mention the EM analysis as supporting evidence for the stiffening of ssDNA.

Minor issues.

1.The formats of the references are not consistent. Some of them have URLs others don't. Please refer to the journal guidelines on the formatting.

We have corrected the reference format.

  1. Figure 1: The caption should be transition dipole moments instead of transition moments. in Figure 1a, it is not clear why the number 3 and 4 are present.

We corrected the figure and figure legend according to the reviewer's comment.

  1. Figure 4: In the caption the number "4" is in a smaller font size than others.

We corrected the font size.

Comments on the Quality of English Language

The English language needs to be improved significantly.

We were surprised by this comment because the text was edited by a professional.

Example:

  1. Abstract: "On the other hand, the signal intensity also depends upon the degree of sample alignment". Here also is redundant. the degree of sample alignment can be changed to the alignment. "Therefore, LD can detect DNA shortening, bending here...." Here LD does not directly detect these effect since LD is not a detection method.

We modified the text.

  1. Principle of Linear Dichroism: "Therefore, if all molecules are aligned in one direction, the absorption of polarized light will be maximum when parallel to the transition moment." The second part of the sentence is misleading, because the author actually meant the absorption will be at maximum when the incident light polarization is parallel to the transition dipole moment of the molecule.

We rewrote the sentence.

"The signal intensity depends, in addition to the chromophore orientation, on the concentration and the absorption coefficient of the chromophore. Therefore, the reduced LD (LDr = LD / isotropic absorbance), is independent of these parameters and conveniently used to analyze the transition moment orientation" The sentence after "Therefore" is not the cause of the sentence before "therefore".

We modified the sentence.

"Since each aromatic chromophore absorbs light at several wavelengths and in a specific plane (several transition moments)" The word Aromatic chromophore is ill-defined and is seldom used in scientific publication. The whole sentence is not clear, although experienced readers can understand what they mean.

We rewrote the sentence.

"the chromophore orientation can be determined by measuring LD at various wavelengths and determining the orientation of all transition moments." In this sentence word "determine" repeated twice and made it confusing on what determines what.

We modified the sentence.

There are more language issues along the way that need to be fixed.

Reviewer 2 Report

Comments and Suggestions for Authors

This review article describe the utility of linear Dichroism measurements for Studying the protein-DNA interactions. After giving a very short introduction, authors have described the principles of LD measurement and given some examples of the studied related to the DNA. Although, a lot of reviews already available regarding the use of linear dichroism for studying the proteins and nucleic acids, a review regarding the protein-DNA interactions was published before (10.3184/003685008X395517) which is also given in the current manuscript as reference # 2 and the current manuscript is somewhat similar to that one. In order to improve the manuscript, I would suggest the authors to compare this technique with more advanced one, i.e., CD spectroscopy because as stated in the manuscript that LD can be performed on the CD spectrophotometer. How LD is more advantageous than CD and how the results will look if a study will performed using both spectroscopies. 

Author Response

This review article describe the utility of linear Dichroism measurements for Studying the protein-DNA interactions. After giving a very short introduction, authors have described the principles of LD measurement and given some examples of the studied related to the DNA. Although, a lot of reviews already available regarding the use of linear dichroism for studying the proteins and nucleic acids, a review regarding the protein-DNA interactions was published before (10.3184/003685008X395517) which is also given in the current manuscript as reference # 2 and the current manuscript is somewhat similar to that one. In order to improve the manuscript, I would suggest the authors to compare this technique with more advanced one, i.e., CD spectroscopy because as stated in the manuscript that LD can be performed on the CD spectrophotometer. How LD is more advantageous than CD and how the results will look if a study will performed using both spectroscopies.

Both LD and CD use polarized light. However, the information obtained by LD differs from that by CD. We add a sentence explaining the difference and complementarity of these measurements.

Reviewer 3 Report

Comments and Suggestions for Authors

The manuscript “Linear Dichroism Measurements for the Study of Protein-DNA Interactions” by Masayuki Takahashi and Bengt Norden provides a review on the use of LD for DNA /protein/ligand binding. By the provided examples it helps to understand the specifics of the method for different cases like intercalation or groove binding for DNA, DNA kinetics e.g. reduction of size by restrictases, DNA bending , ssDNA structural variation for complexation etc. The manuscript has been carefully prepared, it is very well structured, the Figures and references are informative, not too little, not too much, just right. The readability is excellent and the manuscript can be basically published as is.

I would make just a few observation, believe those are just editing and will be corrected in the process. Some of the Figure captions are on two lines. The references formatting may be rechecked as some of those end with a dot, others no. 

Author Response

The manuscript “Linear Dichroism Measurements for the Study of Protein-DNA Interactions” by Masayuki Takahashi and Bengt Norden provides a review on the use of LD for DNA /protein/ligand binding. By the provided examples it helps to understand the specifics of the method for different cases like intercalation or groove binding for DNA, DNA kinetics e.g. reduction of size by restrictases, DNA bending , ssDNA structural variation for complexation etc. The manuscript has been carefully prepared, it is very well structured, the Figures and references are informative, not too little, not too much, just right. The readability is excellent and the manuscript can be basically published as is.

I would make just a few observation, believe those are just editing and will be corrected in the process. Some of the Figure captions are on two lines. The references formatting may be rechecked as some of those end with a dot, others no.

We have corrected the references and the figure legends.

Round 2

Reviewer 1 Report

Comments and Suggestions for Authors

The authors have improved the manuscript and addressed all the issues raised in my previous comment.

Reviewer 2 Report

Comments and Suggestions for Authors

The review is acceptable in its present form.